# Biomolecular Pathways of Cryoinjuries in Low-Temperature Storage for Mammalian Specimens

**DOI:** 10.3390/bioengineering9100545

**Published:** 2022-10-12

**Authors:** Ying Fu, Wenjun Dang, Xiaocong He, Feng Xu, Haishui Huang

**Affiliations:** 1The Key Laboratory of Biomedical Information Engineering of Ministry of Education, Xi’an Jiaotong University, Xi’an 710049, China; 2Bioinspired Engineering and Biomechanics Center (BEBC), Xi’an Jiaotong University, Xi’an 710049, China

**Keywords:** cryopreservation, apoptosis, necroptosis, ischemia-reperfusion injury, molecular basis

## Abstract

Low-temperature preservation could effectively extend in vitro storage of biological materials due to delayed or suspended cellular metabolism and decaying as illustrated by the Arrhenius model. It is widely used as an enabling technology for a variety of biomedical applications such as cell therapeutics, assisted reproductive technologies, organ transplantation, and mRNA medicine. Although the technology to minimize cryoinjuries of mammalian specimens during preservation has been advanced substantially over past decades, mammalian specimens still suffer cryoinjuries under low-temperature conditions. Particularly, the molecular mechanisms underlying cryoinjuries are still evasive, hindering further improvement and development of preservation technologies. In this paper, we systematically recapitulate the molecular cascades of cellular injuries induced by cryopreservation, including apoptosis, necroptosis, ischemia-reperfusion injury (IRI). Therefore, this study not only summarizes the impact of low-temperature preservations on preserved cells and organs on the molecular level, but also provides a molecular basis to reduce cryoinjuries for future exploration of biopreservation methods, materials, and devices.

## 1. Introduction

Biopreservation for mammalian specimens is an enabling technology to bridge the spatiotemporal gap between procurement sources and application destinations for living cells, tissues, organs, and biomacromolecules (such as proteins and mRNA) [1,2,3,4]. Since mammalian specimens can slow down or even suspend their in vitro metabolic activity and decay time at low temperatures, they are usually stored in non-physiological low-temperature conditions to prolong their preservation period [5]. They can be preserved in two states: either at deep cryogenic temperatures in frozen/vitrified states with totally suspended animation for months or even years (cryopreservation usually refers to temperatures from −80 to −196 °C) or at hypothermic temperatures without phase transition with partially decreased animation for hours or days (hypothermic storage around 4 °C) [6,7]. Due to its indispensable role in the supply chain of biological specimens, biopreservation is ubiquitously used in biomedical and clinical research and applications, such as assisted reproduction [8,9,10], stem cell therapy [11,12], regenerative medicine [13], tissue engineering [14], and mRNA medicine [15]. However, mammalian specimens under low-temperature preservation are far from their normal physiological conditions, which inevitably induces biophysical and biochemical changes and injuries to the preserved cells. These injuries are caused not only by the phase transition (e.g., ice formation inside and outside of cells) at sub-zero temperatures during the freezing and retrieving processes [16], but also by the substantially altered states, activities, and functions on the molecular level, which ultimately lead to cell injuries or even death [17,18].

Ice crystals formed during cryopreservation (slow freezing and vitrification) cause mechanical damage to mammalian specimens (e.g., cell deformation, damage to cell and organelle membranes), and the main results of ice crystal propagation are known through the two-factor hypothesis proposed by Mazur [19], which states that ice crystal propagation of mammalian specimens under low-temperature conditions is mainly influenced by the cooling rate and osmotic pressure. The intracellular chemical potential is higher than the extracellular chemical potential during slow cooling when the cell is dehydrated, extracellular water molecules increase, causing extracellular ice crystals to form before intracellular ice crystals. The subsequent increase in extracellular osmotic pressure causes the formation of intracellular ice crystals from unleached water molecules inside the cell. When the cooling rate is too fast, the vitrified state can be reached, and no ice crystals are formed. However, during rewarming, ice crystals are easily generated due to ice recrystallization. To avoid ice crystal damage to biospecimens, common methods include adding cryoprotectants (CPAs), changing the cooling rate, etc. CPAs can be classified according to their source as natural CPAs (e.g., antifreeze proteins (AFPs)) and synthetic CPAs (e.g., polyvinyl alcohol (PVA)) [20]. These CPAs are classified as permeating and nonpermeating types, and a combination of both is often used to reduce toxicity to cells. In addition, cryopreservation technologies can destroy diseased tissues in situ and treat conditions such as vascular lesions in the oral cavity [21], primary tumors [22], and cutaneous lesions [23]. Different preservation technologies are subject to intracellular ice formation (IIF) and solute effects (SEs), but at the molecular level, the cryoinjuries induced by different preservation technologies have unique characteristics. To provide the readers with an intuitive understanding of cryopreservation technologies and corresponding cryoinjuries, the following is a simple description of the main preservation technologies.

According to the characteristics of preservation technologies and preserved specimens, the three main preservation approaches were defined and have been being developed for several decades: slow freezing, vitrification, and hypothermic storage. Slow freezing and vitrification are suitable to preserve biospecimens at the cellular level due to apoptosis and necroptosis, while large tissues and organs are suitable preserved using hypothermic storage due to ischemia–reperfusion injury. For long-term (e.g., >1 month) preservation, slow freezing and vitrification at ultralow temperatures (conventionally named “cryopreservation”) have been commonly adopted [24]. The slow freezing mentioned in this paper is equilibrium freezing. This is because the chemical potential of the cytoplasm of the cell in the supercooled state is higher than that outside of the cell, and in order to ensure the balance of the chemical potential inside and outside the cell so that the cell does not freeze, the cooling rate needs to be slowed down, and for most cells, a slow cooling rate (e.g., 1 °C/min) can significantly reduce the probability of intracellular freezing [19,25], and is widely used to preserve mammalian cells [26,27], microscale cell–biomaterial constructs [28], and micro-tissues [29]. According to the two-factor hypothesis proposed by Mazur, biospecimens undergoing slow freezing are subject to the damages caused by IIF and SEs [30]. During slow freezing, extracellular water is first nuclearized into ice crystals elevating the chemical potential and osmotic pressure of extracellular solutions, inducing an outflow of intracellular water, and eventually leading to cell dehydration. If the cooling rate is high, the intracellular water cannot leave cells before freezing; it ultimately causes substantial IIF and lethal cryogenic damage to the cellular membrane systems and organelles. If the cooling rate is too slow, the cells are subject to severe dehydration and deformation caused by SEs [20]. On the other hand, vitrification utilizes ultrahigh cooling/warming rates (e.g., >10^3^ °C/min) and high CPA concentrations (e.g., 6–8 M) to directly transform liquid biological specimens into a glass-like state without ice formation, avoiding both IIF and SEs in slow freezing [31,32]. As a result, vitrification is generally considered superior to slow freezing for banking stress-sensitive biological samples such as oocytes [33], stem cells [34], and some special tissues [35]. However, the high concentration of cytotoxic CPAs and/or high cooling rates induce significant cytotoxicity, mechanical and chemo-osmotic stresses to specimens, causing cellular viability and functional damage and even cell death.

For short-term (e.g., <1 week) preservation, usually for large tissues and organs, hypothermic storage at around 4 °C (i.e., static cold storage (SCS) with or without hypothermic machine perfusion (HMP)) is mainly adopted [27,36,37]. SCS-preserved organs are flushed with and suspended in an intracellular- (e.g., the University of Wisconsin (UW) preservation solution) or extracellular (e.g., Celsior or Perfadex solution)-like carrier solution on an ice bag [38]. In addition, HMP can continually provide tissues/organs with metabolic substrates with protracted infusion during storage [39]. Although hypothermic storage avoids ice formation, SCS-preserved organs or tissues stored at those unfrozen temperatures still undergo substantial metabolism, usually with hypoxia, nutrient shortage, and metabolic waste accumulation, which ultimately causes ischemic damage and reperfusion injury. Of note, supercooling preservation techniques have recently been developed to store cells, tissues, and organs below 0 °C without freezing [40,41,42]. By removing the primary heterogeneous ice nucleation site of supercooled biospecimens on the water–air interface with surface sealing by an immiscible oil phase, stable deep supercooling preservation was achieved for a deeply supercooled (e.g., ≤−10 °C) specimen without ice formation and CPAs [43]. In addition, freeze drying has become a new method of preservation of mammalian samples (e.g., somatic cells [44], red blood cells [45], sperm [46], and three-dimensional cells [47]). Compared with other preservation technologies, freeze drying requires expensive equipment and technical expertise, but reduces the costs of continuous maintenance of liquid nitrogen and working hours. This also indicates that different mammalian specimens should be preserved in different ways, and researchers need to further define the optimal cryopreservation methods for specific specimens. Furthermore, some other improved cryopreservation techniques exist, such as interrupted slow cooling, two- or three-step methods, liquidus tracking, etc. Interrupted cooling protocols and interrupted rapid cooling preserve mammalian specimens in two steps: first, down to a critical intermediate temperature, and then down to the storage temperature with or without holding time, allowing researchers to investigate the types of cryoinjuries suffered by cells at various stages of cryopreservation. It has been demonstrated that cryoinjuries to cells during the interrupted cooling protocol stage are caused by solute damage [48,49], whereas cryoinjuries to cells during the interrupted rapid cooling stage are caused by ice crystal damage [48]. The two- and three-step methods were improved to protect cells during vitrification from the toxic effects of CPAs. Different cells have different osmotic pressure tolerance limits and toxicity tolerance limits so they need to gradually load or unload CPAs to reduce their toxicity to the cells [50]. The liquidus tracking method reduces the concentration of CPAs and shortens the time cells are exposed to CPAs. This technique solves the problems of CPA toxicity and ice crystal formation during vitrification [51]. The three cryopreservation techniques described above are all improved methods to reduce solute damage, ice damage, and toxicity of CPAs, and the molecular pathways of cryogenic damage to cells under cryogenic conditions are not mentioned. Since the metabolic states involved in the above three methods are similar to those of conventional cryopreservation techniques, we can classify the molecular pathways of cryoinjuries involved in cells in these three preservation techniques into the same category.

Cryoinjuries in biopreservation generally lead to cell death via apoptosis or necroptosis (necrotic apoptosis). Sublethal cell damage in cryopreservation (slow freezing and vitrification) can directly cause apoptosis or necroptosis, while hypothermic storage (HMP and SCS) can indirectly cause apoptosis due to ischemic and reperfusion damage [52,53,54]. Apoptosis is a physiologically programmed cell death in a controlled manner, characterized by the activation of caspases, DNA fragmentation, and membrane blebbing. On the contrary, as a subclass of the necrosis death pathways, necroptosis is characterized by the swelling of cellular organelles and cytoplasm, subsequent rupture of the plasma membrane, and final cell lysis. To suppress cell injuries in cryopreservation, scientists have developed various intracellular and extracellular CPAs, such as dimethyl sulfoxide (DMSO), 1–2 propanediol (PROH), and trehalose, to minimize ice formation, osmotic shock, and excessive cell dehydration and deformation [55,56,57]. In hypothermic storage, although tissues or organs are preserved in intracellular- or extracellular-like carrier solutions [58,59,60], ischemic damage occurs due to the lack of oxygen supply by blood after the tissues/organs are stripped from the donors, and, thereafter, the cellular energy level, ATP concentration, and chelating ability of metabolic wastes decrease substantially, causing ionic imbalance, cell apoptosis and/or necroptosis, and, eventually, cell death. Reperfusion injury, on the other hand, is when the preserved ischemic tissues/organs are reperfused with an excessive oxygen supply, and a large number of reactive oxygen species (ROS, e.g., H_2_O_2_) are produced beyond the chelating capacity of the biospecimens, leading to immune responses (sterile inflammation), apoptosis/necroptosis, and, eventually, cell death [61,62].

Therefore, cell injuries in biopreservation not only result from lethal events such as IIF and osmotic shock, but can also be caused by the development of sublethal damage via apoptosis or necroptosis. Since the physiochemical mechanisms of cryoinjuries (e.g., IIF and SEs) have been summarized elsewhere [27,63], this review focused on the molecular mechanisms of apoptosis, necroptosis, and ischemia–reperfusion injury during low-temperature biopreservation. Therefore, it could provide a molecular basis to regulate and even inhibit the induction and development cascades of cryoinjuries of cells and organs to improve preservation outcomes.

## 2. Morphological Changes in Preservation

Under low-temperature preservation, the morphology of mammalian cells, tissues, or organs could change substantially compared to their physiological conditions (e.g., fragmented cytoplasm, edemas) [64,65,66]. Morphological changes are particularly important for cryopreserved cells since they are closely related to mitochondrial injuries (Figure 1). In a study on sperm cryopreservation, Jamal et al. demonstrated sperm deformity increased significantly post-thawing compared to the fresh group [65]. In addition, Raquel et al. found both fresh control and vitrified hepatocyte spheroids have a regular surface with well-preserved morphology throughout 5 days of culture as demonstrated by SEM imaging, while slow-frozen suspended hepatocytes showed consistently roughed surfaces, suggesting that most cells on the surface of hepatocyte spheroids are either dead or senescent [67]. Men et al. investigated morphological alterations of cryopreserved oocytes, revealing fragmented cytoplasm and apoptotic body formation in oocytes post-vitrification and 3-day culture [68]. Furthermore, Villalba et al. found apoptosis-mediated cell damage and cell loss in meniscus cryopreservation [69]. During the hypothermic storage of tissues/organs, the morphology of the tissues/organs and the function of interior cells could also change significantly. Wan et al. found that hearts preserved at 8 °C showed extensive contraction bands, which is indicative of pathological contracture [70]. Susanne et al. preserved the small bowel at 4 °C, and severe degeneration of enterocytes and leakage of lanthanum into intercellular spaces were observed [71]. Overall, these morphological changes of preserved specimens are manifestations of their cryoinjuries and status post-low-temperature preservation and reflect (at least partially) the molecular cascades that might lead to cell survival or death.

## 3. Apoptotic Pathways of Low-Temperature Biopreservation

Apoptosis is a mechanism of programmed cell death and plays important roles in a variety of organisms, from homeostatic maintenance to the removal of superfluous, aged, or harmful cells [72,73], while necroptosis is caspase-independent cell death in an uncontrolled manner [74]. Apoptosis and necroptosis are different in the aspects of morphological characteristics, molecular responses, and cell consequences. Apoptosis is characterized by chromatin condensation, nuclear fragmentation, formation of apoptotic bodies, activation of intracellular proteases from their inactive zymogens (procaspases to caspases), cellular shrinkage, nonrandom cleavage of DNA into 180 KDa fragments by exonucleases, maintenance of intact cell membranes, and externalization of phosphatidylserine. Particularly, necroptosis features electron-lucent cytoplasm, mitochondrial swelling, random nuclear DNA cleavage, and loss of plasma membrane integrity without drastic morphological changes in the nuclei. Uncontrolled cellular degeneration in necroptosis causes random nuclear DNA cleavage, and leakage of the cytoplasm through plasma membrane disruption induces cellular inflammatory responses.

Apoptosis is a physiological cell death mechanism that occurs in response to environmental and developmental signals, during which dying cells silently vanish without traces left behind. Its central signaling components are a family of cysteine proteases called caspases. Through the activation of caspases, apoptosis can be triggered mainly by two different types of pathways: (1) extrinsic pathways (death receptor-mediated pathways) and (2) intrinsic pathways (mitochondrial pathways) (Figure 2). The extrinsic and intrinsic pathways finally converge to a common execution pathway that involves proteolytic activation of caspases-3 and/or caspases-7 from their inactive procaspases. On the other hand, both extrinsic and intrinsic pathways could be activated in low-temperature biopresrvation as evidenced by the protective roles of selective cysteinase inhibitors (e.g., z-IETD-FMK, z-LEHD-FMK, and z-DEVD-FMK) in preservation [75].

### 3.1. Extrinsic Pathways

The extrinsic pathways of apoptosis are triggered by the binding of death receptors to death ligands (FasL/FasR, TNF-a/TNFR1), which then recruit FADD (Fas-associated death domain) and TRADD (tumor necrosis factor receptor type 1-associated DEATH domain protein, Figure 2). Factor receptor type 1-associated DEATH domain protein binds to procaspase-8 to form a death-inducing signaling complex (DISC), activating caspases and inducing the caspase cascade reaction. In addition, DISC also activates caspase-8 and, thus, cleaves BH3 on the Bid protein to form a truncated Bid (tBID). Then, tBID is transferred to mitochondria, which alters the permeability of the inner mitochondrial membrane and eventually induces the release of cytochrome C, leading to the activation of the intrinsic pathways of apoptosis.

In the apoptotic pathways induced by low temperatures, the activation of extrinsic pathways is mainly caused by caspase-8. Desoutter et al. reported that apoptosis was substantially activated after the thawing of CD3^+^, CD56^+^, and CD14^+^ cells. Since apoptosis is generally associated with intrinsic pathways [76], Bissoyi et al. and Zhang et al. investigated whether a single extrinsic pathway could induce apoptosis [75,77]. The result showed that caspase-8 was activated in the extrinsic pathways, and caspases-3 and -9 were activated in the intrinsic pathways. As shown in Figure 2, the activation of death receptors of extrinsic apoptotic pathways I and II started the caspase cascade reaction, causing a change in mitochondrial membrane permeability and triggering the activation of the intrinsic pathways of apoptosis. These results indicated that apoptosis induced by low temperatures can be activated by extrinsic pathways and converge with intrinsic pathways.

### 3.2. Intrinsic Pathways

Intrinsic, or mitochondrial, pathways of apoptosis involve changes in mitochondrial membrane permeability when cells are stimulated by stresses such as heat, oxidation, and radiation. As mitochondrial membrane permeability increases, proapoptotic protein factors Bax [78,79] enter the mitochondria, which promotes the release of a variety of proapoptotic factors, including cytochrome C, serine protease HtrA2/Omi, and Smac/DIABLO. In addition, ROS generated during cryopreservation also promotes the release of cytochrome C from mitochondria and induces a cascade of caspases to trigger apoptosis [77,80,81]. Therefore, by scavenging ROS, the antioxidant and necrostatin-1 (Nec-1) can significantly reduce mitochondrial swelling and vacuolization, inhibit the release of cytochrome C, prevent caspase-3 activation, and inhibit apoptosis in goat spermatogonial stem cells (SSCs) [54]. Cytoplasmic cytochrome C binds to Apaf-1, activates caspase-9, and generates apoptotic vesicles to activate caspase-3 (Figure 2). On the other hand, Smac/DIABLO activates caspases by suppressing the inhibitors of apoptosis proteins (IAPs) [82,83]. Intrinsic apoptotic pathways III, IV, and V are activated through apoptosis-inducing factors Bax, peroxisomes (e.g., ROS), and changes in mitochondrial membrane permeability, respectively (Figure 2). Particularly, pathway IV is not only an intrinsic pathway, but also an extrinsic pathway through ROCK and p53 to break the nuclear DNA, and eventually leads to apoptosis.

Previous studies showed that apoptosis inhibitors (e.g., caspase inhibitor Z-VAD-FMK [84,85,86] and selective Rho-associated kinase (ROCK) inhibitor Y-27632 [87]) can effectively improve cellular hypothermia tolerance ability. Surojit et al. used the broad-spectrum cysteinase inhibitor z-VAD to investigate the mechanism of cryo-induced apoptosis in peripheral blood mononuclear cells (PBMCs) [88]. They not only demonstrated the ability of z-VAD to inhibit mitochondrial membrane perturbation and apoptotic cell death, but also indicated that cysteinase-mediated mitochondrial membrane damage is associated with freeze-induced apoptosis of CD4^+^ T cells. In addition, when stallion spermatozoa were frozen using inhibitors of the mitochondrial permeability transition pore (MPTP), a significant decrease in cysteinase activity and membrane permeability and an increase in the mitochondrial membrane potential were observed in the spermatozoa post-thawing [89]. Therefore, it can be concluded that low-temperature preservation can induce apoptosis through intrinsic/mitochondrial pathways.

Since low-temperature preservation activates apoptotic pathways via multiple pathways (Table 1), various inhibitors of these pathways could be utilized to prevent apoptosis. Maria et al. demonstrated the usage of Z-VAD-FMK during vitrification/warming and post-warming culture can partially inhibit cryopreservation-induced apoptosis by reducing the level of active caspase-3, which improves cryotolerance of bovine in vitro produced (IVP) embryos [84]. Glycerol is a widely used CPA in cryopreservation with an excellent antiapoptotic capability, and Zeng et al. demonstrated that 2% and 3% glycerol have the best antiapoptotic effects for boar spermatozoa [90]. However, the mechanism and relationships between glycerol and apoptosis-related gene expressions need to be further clarified. Furthermore, osmotic shock, oxidative stress, and ROS generation in biopreservation aggravate cryodamage [91,92]. High concentrations of ROS deplete ATP and cause apoptosis and cell death [93]; thus, antioxidants can scavenge the released free radicals and, thus, reduce oxidative stress and inhibit the related apoptotic pathways. Mahin et al. utilized vitamin D as an antioxidant in a cryopreservation solution to improve the viability and functionality of human sperm [94].

## 4. Necroptotic Pathways of Low-Temperature Biopreservation

Necroptosis is the characterized programmed necrosis, which is triggered by cytokines and pattern recognition receptors (PRRs) and is modulated in a RIPK1- and RIPK3-dependent manner and also involves the tumor necrosis factor (TNF) signaling pathway [97,98]. Previous studies demonstrated that low-temperature preservation causes cell death via programmed necrosis, i.e., necroptosis (Table 2), though the exact molecular mechanisms have not been fully elucidated [54,99]. It has been shown that the concentration of intracellular tumor necrosis factor (TNF-α) increases after cryopreservation in both peripheral blood mononuclear cells [100] and pancreatic islets [101], suggesting a TNF-α-mediated pathway of cryopreservation-induced cell death. Therefore, cell necroptosis could be another important molecular mechanism of cryoinjuries in low-temperature preservation.

### 4.1. Initiation of Necroptosis: TNFR1 Decides

As a member of the tumor necrosis factor family, TNF-α is also involved in the necroptosis pathways induced by low-temperature preservation. The binding of TNF-α to the death receptor TNFR results in a conformational change of TNFR, which recruits multiple proteins (i.e., TNFR1-associated death domain protein (TRADD), RIPK1, TRAF2, E3 ubiquitin ligase, cIAP1/2, and LUBAC) to form TRADD- and RIPK-dependent complex I (Figure 3). This complex I forms a TGF-activated kinase 1 (TAK1)-binding protein (TAB) complex by recruiting the TGF-activated kinase 1 (TAK1)-binding protein (TAB) complex, which could transduce proinflammatory and prosurvival signals to activate nuclear factor κB (NF-κB) signaling [105]. When RIP1 is poly-ubiquitinated, complex I gets unstable, dissociates RIPK1, and forms complex II by interacting with TRADD, FADD, procaspase-8, and FLIP (Figure 3) [106]. Oligomerization and phosphorylation of RIPK3 in necrosomes lead to the recruitment and phosphorylation of MLKL, which then translocates to the plasma membrane and leads to membrane damage and necrosis [98,106].

### 4.2. Initiation of Necroptosis: ROS Decides

Ischemia and reperfusion in the hypothermic storage of large tissues and organs also trigger necroptosis. The shortage of oxygen and nutrients in ischemia not only causes cells to consume the ATP reserve, but also decreases the efficiency of ATP production, causing the reversal of the electron transport chain (ETC) in the mitochondrial inner membrane and the accumulation of electron carriers, such as succinate. As a result, abrupt supply of oxygen upon reperfusion allows the mitochondria to produce excessive ROS that cannot be scavenged and removed promptly (Figure 3). Furthermore, this overloaded ROS could also lead to DNA damage, excessive activation of poly (ADP-ribose) polymerase (PARP), and final necroptosis [107]. In addition, the same mechanism as in the ischemic injury state (Figure 4) can be triggered. Cellular ATP concentration and energy level decrease in hypothermic storage, and the Na^+^/K^+^ ATPase is unable to pump out the excess Na^+^, causing cells to load extracellular Ca^2+^ by reversing the Na^+^/Ca^2+^ transporter protein [108]. Intracytoplasmic Ca^2+^ concentration can also be elevated by Ca^2+^ release from the endoplasmic reticulum (ER) [109]. When the intracytoplasmic Ca^2+^ content rises to a certain high level, the Na^+^/Ca^2+^ reverse transporter protein stops transferring Ca^2+^ from the mitochondria, causing the mitochondrial Ca^2+^ level to rise beyond the safe limit.

## 5. Molecular Pathways of Ischemia–Reperfusion Injuries

Although acute myocardial infarction, stroke, and spinal cord injury are associated with apoptosis and inflammation caused by IRI to the heart, brain, and spinal cord, the IRI discussed in this section is caused by low-temperature preservation of large tissues and organs. Large tissues and organs are typically preserved at low temperatures (around 4 °C) with reduced but not arrested metabolism and decaying because spontaneous ice formation in cryopreservation is almost inevitable in large tissues and organs at deep cryogenic temperatures (e.g., −80 °C), causing lethal damage to their complex hierarchical structures. Ischemia–reperfusion injury (IRI) can be introduced during tissue/organ acquisition, preservation, and transplantation, impairing their viability and functions. On the other hand, the tissues and organs are preserved with oxygenated solutions via continuous machine perfusion; they do not enter an ischemic state and suffer from IRI. IRI is a multifactorial non-antigen-dependent inflammatory state that can severely compromise early- and long-term functions of allografts. Ischemia results in insufficient oxygen and nutrient supply, causes hypoxia, and primes tissues/organs for subsequent reperfusion injury [110,111]. Organ ischemia can be divided into warm ischemia (WI) and cold ischemia (CI). The former refers to transient ischemia during artery entrapment and excision, and the latter occurs during organ graft preservation at low temperatures. In addition, since there are multiple types of cells within tissues/organs (e.g., hepatic sinusoidal endothelial cells, hepatocytes, Kupffer cells, neutrophils, and platelets in liver) and they could have distinct responses to IRI (WI causes damage mainly to stem cells, CI—to hepatic sinusoidal endothelial cells) [112], the cellular and molecular mechanisms that regulate IRI form a complex network of interconnected molecular pathways [113]. On the other hand, tissues and organs are preserved with oxygenated solutions via continuous machine perfusion; they do not enter an ischemic state and suffer from IRI.

### 5.1. Ischemia Injury

During hypothermic static storage, cells could undergo anaerobic respiration in ischemic conditions, resulting in the production of lactic acid with a significant decrease in pH and a risk of acidosis. Although small amounts of lactic acid can be alleviated by a Na^+^/H^+^ pump on the cell membrane, excessive accumulation of lactic acid increases the intracellular Na^+^ concentration and membrane Na^+^/Ca^2+^ exchange activity (Figure 4). As a result, the concentration of intracellular Na+ and Ca^2+^ elevates and causes intracellular hypernatremia [114]. When intracellular Ca^2+^ levels exceed the cellular clearance load, it triggers the second lethal injury of organ apoptosis, peroxide ROS overload. In addition, Ca^2+^ concentration can be further increased due to the release from the endoplasmic reticulum after mitochondrial damage, causing excessive ROS production and ultimately leading to cellular damage [109,115]. On the other hand, the ischemic injury decreases its energy level as insufficient oxygen supply leads to anaerobic metabolism, activity suppression of electron transport chain complexes (e.g., complexes I–IV), disruption of the electron transport chain, and a decrease in ATP depletion [116]. Since intracellularly accumulated Na+ cannot be removed by ineffective Na/K ATPases, it leads the Na^+^/Ca^2+^ exchanger to start working in the reverse direction, aggravating the issue of intracellular Ca^2+^ overloading and lipid peroxidation to trigger apoptosis [117]. Of note, it was found that IRI during hypothermic preservation can be mitigated by the presence of Zn by regulating the opening of MPTP, reducing ROS production, and preventing lipid peroxidation [118,119].

### 5.2. Reperfusion Injury

Once oxygen supply is restored upon reperfusion, the damage to ischemic tissues/organs, surprisingly, is only aggravated. This is because the availability of oxygen in the reperfusion process leads to an abrupt increase in ROS, which exceeds the scavenging capacity of the preserved tissues/organs. The overwhelming production of ROS destroys proteins, lipids, DNA, cell organelles, and cellular homeostasis. For example, an ischemic and reperfused kidney induces intracellular Ca^2+^ accumulation, renal adenosine activation, and alterations of the superoxy-induced membrane to cause severe renal IRI [120]. In addition, lipid peroxidation, protein and DNA oxidative damage upon reperfusion also aggravate kidney IRI [119]. Therefore, hypothermic storage and subsequent reperfusion inevitably cause kidney injuries, leading to acute renal failure [121].

For the liver, endothelial cells of the liver sinusoids are located in the intima of blood vessels and deliver nutrients and oxygen to hepatocytes. Upon reperfusion, liver damage increases the number of cell adhesion molecules (e.g., P-selectins, E-selectins) on hepatic sinusoidal endothelial cells in hypothermic preserved livers [122]. On the other hand, damaged hepatic sinusoidal cells and hepatocytes release series of inflammatory cytokines, including IL-1β, IL-6, TNF-α, TGF-β, and DAMP, while DAMP further induces Kupffer cells to produce IL-1-β, TNF-α, IFN-γ, and IL-12, which ultimately triggers the inflammatory response of hepatocytes [123,124]. Then, neutrophils enter hepatocytes and Kupffer cells and secrete chemokines (e.g., CXCL1 and CXCL2), creating a chemokine gradient and bringing more neutrophils into the hepatic sinusoidal endothelium via chemokine-driven chemotaxis. Consequently, damaged hepatocytes release HMBG-1 and DNA to induce neutrophils to release ROS (Figure 4) [113,125,126].

Particularly, the small intestine is highly sensitive to IRI due to its high expression of intestinal histocompatibility antigens, the presence of a large number of resident immune cells, microbial colonization, and strong innate immunity. This is also one of the main reasons for the limited development of small intestine transplantation [127]. Gastric IRI generates a large amount of ROS such as superoxide, hydrogen peroxide, and hydroxyl radicals and, thus, is usually associated with high morbidity and mortality of hemorrhagic shock, peptic ulcer, and ischemic gastrointestinal diseases [128].

## 6. Summary

Although low-temperature preservation of mammalian specimens provides great convenience for scientific research and clinical applications, the unphysiological, even harsh, in vitro preservation conditions and procedures subject biospecimens to lethal and sublethal events. The challenges of biopreservation are often associated with ice formation and CPA toxicity, while the molecular basis of the impact of low-temperature conditions is often ignored. Currently, tackling these cryoinjuries requires approaches such as adding apoptosis inhibitors, necroptosis inhibitors, and ROS inhibitors during freezing or recovery procedures. Although progress has been made during the past two decades, some major challenges remain to be solved. 

Here, we summarized the molecular mechanisms of induction and development of cryogenic damage in low-temperature preservation of mammalian biospecimens, including apoptosis, necroptosis, and IRI. Particularly, mitochondria play a central role in the induction and development of cryoinjuries in all kinds of low-temperature preservation methods and, thus, can be generally regarded as an intervention target to improve preservation outcomes [129,130]. Overall, these molecular mechanisms of cryoinjuries in biopreservation provide general guidance to discover cryoprotective agents, design storage solutions, and develop preservation protocols. As such, many research opportunities can be uncovered to improve the low-temperature biopreservation from the molecular perspective.

## Figures and Tables

**Figure 1 bioengineering-09-00545-f001:**
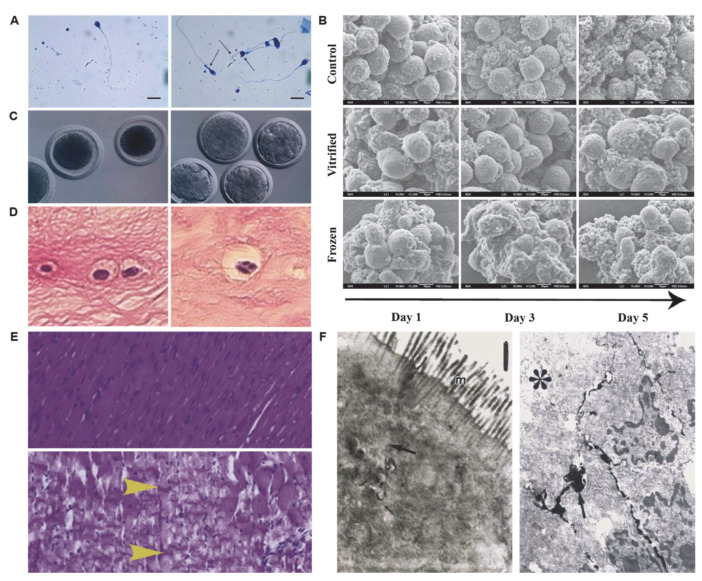
The morphology of cryopreserved cells. (**A**) Representative morphology of fresh (left) and cryopreserved (right) sperm. The arrows indicate abnormal morphology. Scale bars are 10 µm. Adapted with permission from Ref. [65]. Copyright 2021, Elsevier. (**B**) SEM images taken on Days 1, 3 m and 5 of control, vitrified, and frozen HSs. Adapted with permission from Ref. [67]. Copyright 2012, Elsevier. (**C**) Detection of apoptosis in cryopreserved bovine oocytes by morphological characterization and biochemical characterization. Adapted with permission from Ref. [68]. Copyright 2003, Elsevier. (**D**) The morphology of fibrochondrocytes in fresh and cryopreserved meniscus. Adapted with permission from Ref. [69]. Copyright 2011, Springer Nature. (**E**) Hematoxylin and eosin stains of the tissue taken from the left ventricle of healthy hearts or preserved under a variety of conditions. Adapted with permission from Ref. [70]. Copyright 2018, Elsevier. (**F**) Photomicrographs at the EM level of the small bowel of a rat that was not stored or stored for 12 h with RL solution. Adapted with permission from Ref. [71]. Copyright 2004, Elsevier.

**Figure 2 bioengineering-09-00545-f002:**
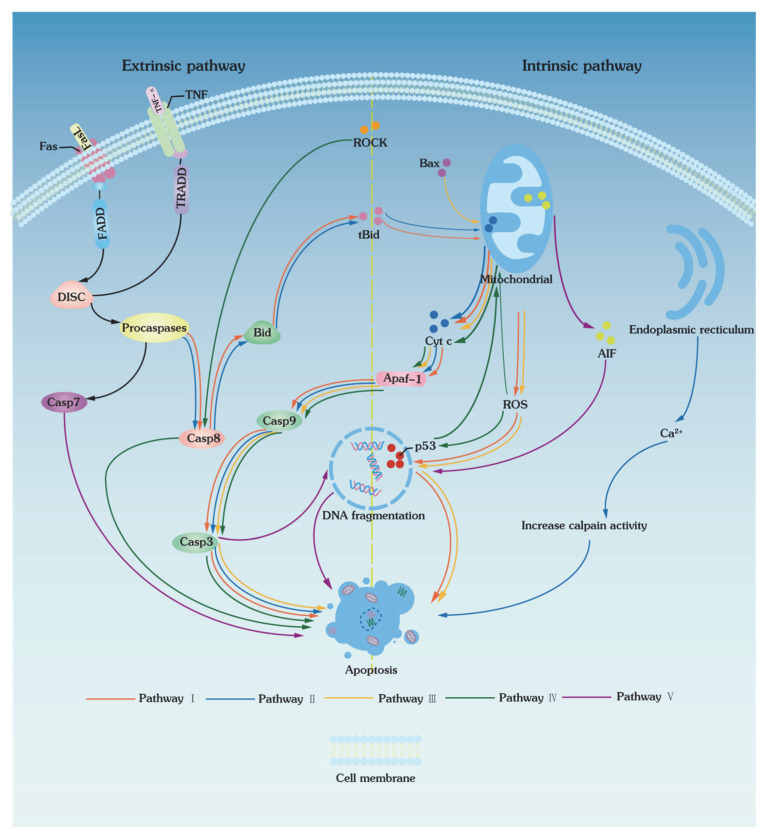
Proposed hypothetical model of cryopreservation-induced apoptosis via caspases. Pathways I and II are extrinsic pathways that stimulate the intrinsic pathways, while pathways III, IV, and V are intrinsic pathway that stimulate the extrinsic pathways. Cryopreservation causing cells apoptosis includes extrinsic and intrinsic pathways; there are five apoptotic pathways summarized based on the previous studies. In the extrinsic pathways, ROCK and procaspases are activated, which induce the caspase cascade reaction. In the intrinsic pathways, the release of Bax, tBid, and ROS induces the mitochondrial permeability transition pore to be opened, resulting in a caspase cascade reaction or DNA fragmentation to trigger apoptosis.

**Figure 3 bioengineering-09-00545-f003:**
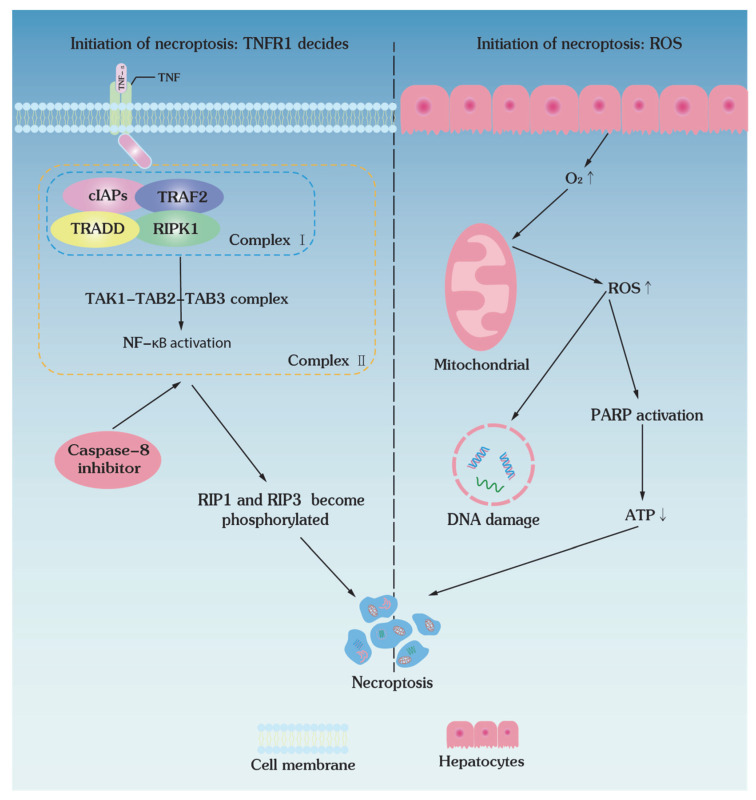
Mechanisms of necroptosis in cells and organs. In mammalian specimens at low-temperature conditions, cellular complex I is formed by TRADD, RIPK1, TRAF2, E3 ubiquitin ligase in cells and activates NF-κB by recruiting TAB. Due to the dissociation of complex I mediated by caspase-8 inhibitors, RIPK3 oligomerization and phosphorylation result in cell necroptosis; during ischemia–reperfusion injury, organs produce Ca^2+^ and ROS, which cannot be cleared. Overloaded ROS activates PARP and leads to cell necroptosis.

**Figure 4 bioengineering-09-00545-f004:**
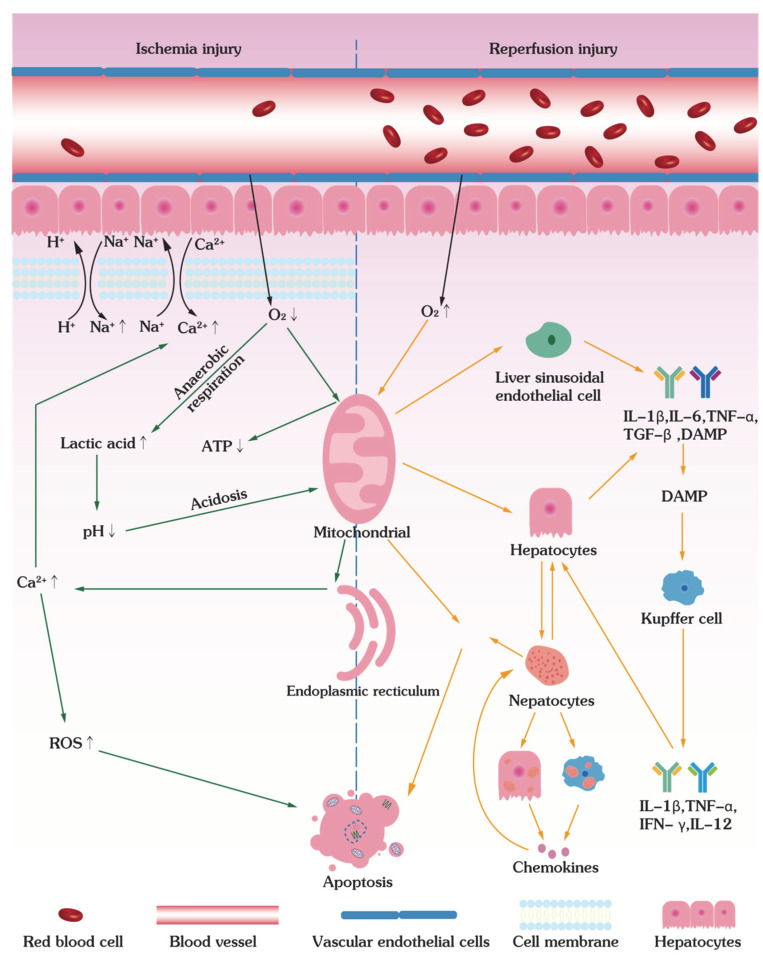
Mechanisms involved in hepatic ischemia–reperfusion injury. The figure on the left illustrates the mechanism of ischemic injury: intracellular PH decreases due to the lack of oxygen, acidosis, and hypoxia jointly lead to acute mitochondrial damage. The increase in intracellular Ca^2+^ also increases ROS. The right side shows a reperfusion injury mechanism: the number of molecules attached to hepatic sinusoidal endothelial cells increases acutely; they are damaged and release various inflammatory factors, leading to various inflammatory responses in hepatic cells and an increase in ROS levels that eventually leads to cell apoptosis.

**Table 1 bioengineering-09-00545-t001:** Low temperatures activate apoptotic pathways.

Study	Species	Cell Type	Cryopreservation Method	Apoptotic Pathway
Jian-Min et al. [77]	Rats	Granulosa cells	Vitrification	Pathway I
Xiangyun et al. [87]	Humans	Embryonic stem cells	Slow freezing	Pathway IV
Akalabya et al. [75]	Humans	Mesenchymal stem cells	Slow freezing	Pathway II
Xia et al. [95]	Humans	Embryonic stem cells	Slow freezing	Pathway IV
Noorollah et al. [96]	Humans	Sperm	Vitrification	Pathway III
Surojit et al. [88]	Rhesus macaques	Peripheral blood mononuclear cells	Slow freezing	Pathway V
Mélanie et al. [52]	Rats	Hepatocytes	Slow freezing	Pathway III
Laboratoire et al. [76]	Humans	Hematopoietic stem cells	Slow freezing	Pathway I

**Table 2 bioengineering-09-00545-t002:** Low temperatures trigger necroptosis.

Study	Species	Cell or Organ Type	Preservation Method
Xie et al. [102]	Mice	Spermatogonial stem cells	Slow freezing
Venkataraman et al. [100]	Humans	Peripheral blood mononuclear cells	Slow freezing
Vara et al. [101]	Humans	Islets	Slow freezing
Dmitriev et al. [103]	Rats	Heart	Static cold storage
Kim et al. [104]	Humans	Bronchial epithelium	Hypothermic machine perfusion

## Data Availability

Not applicable.

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
