# Peer review of "Biomolecular Pathways of Cryoinjuries in Low-Temperature Storage for Mammalian Specimens"

_bioengineering, 2022, doi:10.3390/bioengineering9100545_

Round 1

Reviewer 1 Report

Dear authors

This interesting manuscript has studied different types of cryoinjuries and molecular changes during cryopreservation of the mammalian bio-samples.

The title cab be changed with a better correlation to aim of the study.

The contents of the manuscript has been well categorized but introduction can be improved and oriented to "cryoinjuries" not cryopreservation protocols.

Please pay more attention to the word "pathway" or "pathways". It must correct in several parts of the MS. You can find some highlighted examples within the attached MS. 

The English writing is moderate and needs consideration.

Best regards

Fathi

Author Response

Response to Reviewer #1

Preservation technologies play an important role in biomedical applications, but we cannot ignore the various cryoinjuries caused to biological samples due to low-temperatures. In the preface, we combined preservation techniques with cryoinjuries so that readers can visualize the different cryoinjuries caused by different preservation techniques. Thank you very much for your suggestions, and we have made changes in accordance with your suggestions, which are shown in detail as follows.

Comment #1. The title cab be changed with a better correlation to aim of the study.

Response: We thank the reviewer for this suggestion. We tried to change the title to correlate with aim of the study.

“Biomolecular Pathways of Cryoinjuries in Low-temperature Storage for Mammalian Specimens”

Comment #2. The contents of the manuscript has been well categorized but introduction can be improved and oriented to "cryoinjuries" not cryopreservation protocols.

Response: We thank the reviewer for this suggestion. The cryoinjuries are inseparable from the cryopreservation protocols. In order to let readers visualize the cryoinjuries, the cryopreservation protocols are described together with the cryoinjuries. Different preservation technologies are subject to intracellular ice formation (IIF) and solute effects (SE), but at the molecular level, the cryoinjuries induced by different preservation methods has its own characteristics. To provide the reader with an intuitive understanding of preservation technologies and their induced cryoinjuries, the following is a simply description of the main preservation technologies. Slow freezing and vitrification to preserve biological specimens at the cellular level is highly susceptible to apoptosis and necrosis, while large tissues and organs are highly susceptible to ischemia-reperfusion injury when they are preserved using hypothermic storage. To clarify this, we have explained this by adding following sentences in the highlighted section in the introduction (page 4 of the revised manuscript, lines 79-84, and lines 88-91).

“Different preservation technologies are subject to intracellular ice formation (IIF) and solute effects (SE), but at the molecular level, the cryoinjuries induced by different preservation methods has its own characteristics. In order to provide the reader with an intuitive understanding of preservation technologies and their induced cryoinjuries, the following is a simply description of the main preservation technologies.”

“Slow freezing and vitrification to preserve biological specimens at the cellular level is highly susceptible to apoptosis and necrosis, while large tissues and organs are highly susceptible to ischemia-reperfusion injury when they are preserved using hypothermic storage.”

Comment #3. Please pay more attention to the word "pathway" or "pathways". It must correct in several parts of the MS. You can find some highlighted examples within the attached MS. 

Response: We thank the reviewer for this suggestion. We have completely amend these incongruences. Please see line 248, line 258, line 265, line 268, line 269, line 270, line 272, and line 335 of the revised manuscript.

“line 248, line 258, line 265, line 268, line 269, line 270, line 272, and line 335.”

Comment #4. The English writing is moderate and needs consideration. 

Response: We strongly agree with the reviewer’s concern on our English writing. We apologize for the confusing English writing and this revised manuscript has been professionally edited to ensure accuracy and clarity.

Reviewer 2 Report

The manuscript “Molecular mechanisms of cryoinjuries in low-temperature preservations for mammalian specimens” is devoted to giving knowledge about how to occur cryoinjuries by low-temperature preservations. The review is done concisely and soundly, although the presentation is somewhat dryish and formal. Anyway, I read with interest this manuscript concerning the study of cryoinjuries in low-temperature preservations.

Some points have to be corrected.

Major points

1. It will be a better review if you add information about recent trends, such as the method to overcome cryoinjuries in low-temperature preservation. 

2. Recently, Wakayama S. et al. reported freeze-dried somatic cells can produce healthy, suggesting advantages of their technique, such as being cheaper and safer without liquid nitrogen-free biobanking solutions. What is your opinion concerning this? Are you sure low-temperature preservation is a useful method for mammalian specimens?

Minor points

1. Line 65: Is liquidous correct? If it is liquidus, please amend. 

2. Line 139: Please change “survive” to “survival”.

3. Line 178 and Line 231-232: Please check the word “cysteinase”. Is it right? Is the correct word “caspase-3”?

4. Line 285: Please insert “are” between “specimens” and “at”.

5. Line 364: Please change “injuries to kidney” to “kidney injuries”.

6. Unfortunately, it is a little hard to read the article because many words are getting off. If possible, could you reduce the number of them? 

Author Response

Response to Reviewer #2

The question of how to solve this problem regarding cryoinjuries is an urgent issue in the field today. From the improvement of preservation techniques to the discovery of low-temperatures induced apoptosis, necroptosis and IRI, researchers have tried various antioxidants (e.g. melatonin, resveratrol, ascorbic acid and ROS inhibitors) and apoptosis inhibitors, but have failed to address the upstream targets of hypothermia injury. The role of mitochondria in hypothermic environments has received much attention in recent years, but whether mitochondria play a decisive role in hypothermic injury remains to be investigated. Thank you very much for your suggestions, and we have made changes in accordance with your suggestions, which are shown in detail as follows.

Major points:

Comment #1. It will be a better review if you add information about recent trends, such as the method to overcome cryoinjuries in low-temperature preservation.

Response: We thank the reviewer for this suggestion. This is a key issue the one that we could not ignore in the process of writing the manuscript. First, for cryoinjuries whose molecular mechanisms have been illustrated, i.e. apoptosis or necroptosis, the corresponding solution is to add antioxidants1-4, apoptosis inhibitors5, necroptosis inhibitors6 during freezing or retrieving processes. Second, the specific mechanisms of many cryoinjuries biospecimens during preservation have not been elucidated, so specific methods to overcome cryogenic damage are not known to date. However, in recent years, more and more studies have shown that mitochondria play an indispensable role at low-temperatures7,8, as explained in the summary section (the last paragraph) and will not be repeated here. To clarify this, we have added following sentences in the highlighted (second to the last paragraph) summary section.

“The current methods for solve the cryoinjuries (apoptosis, necroptosis, and IRI) include add apoptosis inhibitors, necroptosis inhibitors, and ROS inhibitors during freezing or retrieving processes. Although progress has been made during past two decades, some major challenges remain to be solved.”

Comment #2. Recently, Wakayama S. et al. reported freeze-dried somatic cells can produce healthy, suggesting advantages of their technique, such as being cheaper and safer without liquid nitrogen-free biobanking solutions. What is your opinion concerning this?

Response: We thank the reviewer for this suggestion. We have read the reviewer's recommendation and acknowledge the innovation of using freeze-drying for somatic cell preservation, which provides a new cryopreservation method for mammalian specimens preservation. Freeze-drying is widely used for the storage and transportation of drugs9 and vaccines10, saving a lot of transportation and storage costs. In recent years, researchers have tried to use this method for the preservation of mammalian specimens, but the preservation effect is not satisfactory, and there is less research on the cryoinjuries induced by this technology11. What is known is that the cryoinjuries involved in freeze-drying includes solute damage, mechanical damage and oxidative damage12, among which it is not known whether the oxidative damage is consistent with the mechanism of cryogenic damage induced by conventional cryopreservation techniques. To clarify this, we have added following sentences in the highlighted in the introduction (page 5 and 6 of the revised manuscript, lines 137-142).

“In addition, freeze-drying of somatic cells has become a new method for mammalian samples. Compared with other methods, freeze-drying holds the promise of being the most convenient, safe and inexpensive cryopreservation method for biological preservation. This also indicates that different species of mammalian specimens are preserved in different ways, and researchers need to further delineate the cryopreservation methods for different species of mammalian specimens.”

Comment #3. Are you sure low-temperature preservation is a useful method for mammalian specimens?

Response: We thank the reviewer for this suggestion. Low-temperature preservation is to store biological tissues (i.e. mammalian specimens) at low temperatures to delay or suspend their metabolic activity and decaying rates so that they can be preserved for extended periods for future use. As discussed in this paper, it includes three major approached, slow-freezing, vitrification, and hypothermic storage. However, since there are many different types of mammalian specimens with distinct characters, specimen-specific preservation methods and conditions would be required to obtain optimal preservation outcomes. For example, germ cells such as oocyte13, sperm14, etc. would be best preserved using vitrification. Red blood cells, large-volume tissues, and organs need to be preserved by hypothermic storage (usually 4℃) without ice formation to ensure maximum efficiency15,16. All these facts suggest that the cryopreservation of mammalian specimens still requires continuous investigation to find the optimal preservation technologies for the preservation of samples. To clarify this, we have added following sentences in the highlighted in the summary section (page 6 of the revised manuscript, lines 140-142).

“This also indicates that different species of mammalian specimens are preserved in different ways, and researchers need to further delineate the cryopreservation methods for different species of mammalian specimens.”

Minor points:

Comment #4. Line 65: Is liquidous correct? If it is liquidus, please amend. 

Response: We thank the reviewer for this suggestion. We have changed from liquidous to liquidus in the manuscript though both forms can be found in the documents.

“Line 113.”

Comment #5. Line 139: Please change survive to survival.

Response: We thank the reviewer for this suggestion. We are sorry for this error and have fixed it.

“Line 218.”

Comment #6. Line 178 and Line 231-232: Please check the word cysteinase. Is it right? Is the correct word caspase-3

Response: We thank the reviewer for this suggestion. After verification, the wording of the above two words is correct.

Comment #7. Line 285: Please insert are between specimens and at.

Response: We thank the reviewer for this suggestion. We are sorry for this error and have fixed it.

Comment #8.  Line 364: Please change injuries to kidney to kidney injuries.

Response: We thank the reviewer for this suggestion. We are sorry for this error and have fixed it.

“Line 426.”

Comment #9. Unfortunately, it is a little hard to read the article because many words are getting off. If possible, could you reduce the number of them? 

Response: We thank the reviewer for this suggestion. We try to reduce their number according to your suggestions

Reviewer 3 Report

General comment: 

This manuscript reviews the relationship between cryopreservation and molecular mechanisms of cell death, namely apoptosis, necroptosis and necrosis.  It briefly introduces the basics of cryopreservation schemes that include slow cooling (which would be more accurately described as equilibrium or quasi equilibrium cooling), vitrification (which is incorrectly identified with only “ultra-rapid” cooling protocols), and a somewhat imprecisely defined hypothermic preservation regime that leaves out much of the recent work on high-subzero preservation approaches. The manuscript is good for identifying and reviewing these mechanisms of cell death, and covers a decent amount on sub-physiologic but above zero temperature regimes.  The review does a good job covering the relationship between ischemia reperfusion injury and these molecular mechanisms. However, the specifics of how they are connected directly to cryopreservation is not well covered. Is it that cells are predisposed to this injury due to cryo-induced stressors? Is not IRI associated with all transplantations? The paper would be much more relevant to cryobiological audiences if some more thought was placed into these types of concerns. The relationships between the molecular mechanisms and cryopreservation induced injuries is not well explored, however, at lower, subzero, temperature regimes. There is no discussion about the relationship between ice formation and these mechanisms, there is no discussion about the relationship between cryoprotectant equilibration induced damage and these mechanisms, whether due to chemical toxicity or to the associated volume changes. There is no discussion about the relationship of thermal stress induced during vitrification of tissues, nor the vast literature on cryosurgery and these mechanisms. 

In short, I find this a fair but somewhat shallow introduction to this specific intersection of the literature. To me it does not contribute much novel synthesis and leaves a lot out that should be included on a number of relevant topics. A major revision should address CPAs, ice formation, mechanobiological concerns, the cryosurgery literature, at the least. 

More specific comments: 

Respectfully, while the manuscript has clear English with only a few cases where the meaning is obscured, there are a number of grammatical and typographical concerns with this article that should be addressed before publication. As this reviewer has asked for a number of major revisions, we will not go through any specific instances here. 

As noted above, “slow cooling” is inaccurate, please use equilibrium and/or quasiequlibrium cooling. This is because organs and large tissues are vitrified using slow cooling approaches. Along these lines, “vitrification” as defined in the manuscript only includes high concentration/high cooling rates. Again, organs/large tissues are vitrified using slow cooling rates, and note that water can be vitrified without any cryoprotectant. Finally, the authors leave out some of the “middle ground” but successful techniques including interrupted slow cooling, two- or three-step methods, liquidus tracking, etc. 

Please ensure that all abbreviations are defined in the first instance. 

pH should be capitalized as such, Ca2+ should be superscripted throughout. 

Figure 4 is unclear. What are the different layers, lines, cell types? 

There is an enormous literature on sperm cryopreservation and antioxidants, and a more recent literature on somatic cell and tissues and antioxidants in cryopreservation. The authors should include a more thorough investigation of these applications and their relevance to the molecular mechanisms. Examples include ovarian cells, ovarian tissues, testicular tissues, and bull, sheep, etc sperm, and more. 

In general, it would be helpful for the reader to have the species as well as the cell type when examples are given as there seem to be large differences in cryo-response within cells but among species. Look at oocytes and sperm for basic examples. 

Author Response

Many excellent papers have been published on the knowledge of cryoprotectants, ice crystal formation, and heat transfer during preservation involved in cryopreservation, which are closely related but not coincident with this paper. Particularly, we highlight the biomolecular mechanisms of cryogenically induced damages. Thank you very much for your suggestions, the detailed explanations are provided as follows.

Comment #1. Is it that cells are predisposed to this injury due to cryo-induced stressors?

Response: We thank the reviewer for this suggestion. A large number of studies have shown that when cells are cryopreserved, the concentration of intracellular pro-apoptotic factors and tumor necrosis factors increases, and combined with the apoptotic and necroptosis phenomena exhibited by cells after low-temperature preservation17,18, we agree that the apoptotic and necroptosis pathways are stimulated by the cells under low temperature conditions.

Comment #2. Is not IRI associated with all transplantations?

Response: We thank the reviewer for this suggestion. Acute myocardial infarction19, ischemic stroke20, and spinal cord injury21 are associated with IRI to the heart, brain, and spinal cord. IRI in our manuscript are all described in the context of hypothermia (organ low-temperature preservation and retrieving processes), so we have not mentioned them except for hypothermia-induced IRI. However, if the organs are preserved by machine perfusion, rather than hypothermic storage, the organ transplantation would not suffer from IRI, since the organs are constantly perfused with oxygenated solution and do not enter ischemic state. To clarify this, we have added the logical flow in the highlighted section 4. Please see page 13 of the revised manuscript, lines 367-370, lines 376-378.

“Although acute myocardial infarction, stroke, and spinal cord injury are associated with apoptosis and inflammation caused by IRI to the heart, brain, and spinal cord, the IRI discussed in this section is caused by the low-temperature preservation of large tissues and organs.”

“On the other hand, the tissues and organs are preserved with oxygenated solutions via continually machine perfusion, they would not enter an ischemic state and suffer from IRI.”

Comment #3. To me it does not contribute much novel synthesis and leaves a lot out that should be included on a number of relevant topics. A major revision should address CPAs, ice formation, mechanobiological concerns, the cryosurgery literature, at the least. 

Response: We thank the reviewer for this suggestion. As indicated by the introduction, this paper aims to give the reader a comprehensive understanding of molecular mechanism of cryoinjuries caused by different cryopreservation techniques. We do not focus on the aspects of CPA, ice formation, mechanobiological concerns, and cryosurgery, which have been reviewed elsewhere or researchers have not obtained much understandings, though they are closely relevant to the success to cryopreservation. Nevertheless, we mentioned and discussed some important perspectives on their effects as demonstrated in page 3, lines 60-79.

“Ice formed during cryopreservation (slow freezing and vitrification) cause mechanical damage to mammalian specimens (e.g. cell deformation, damage to cell and organelle membranes), and the main factors of icing are known according to the two-factor hypothesis proposed by Mazur19, which states that icing of mammalian specimens under low temperature conditions is mainly influenced by the cooling rate and osmotic pressure. During slow cooling, the intracellular chemical potential is higher than the extracellular chemical potential, when the cell is dehydrated and extracellular water molecules increase to generate extracellular ice crystals before the intracellular ones. The subsequent increase in extracellular osmotic pressure leads to the formation of intracellular ice crystals from unleached water molecules inside the cell. When the cooling rate is too fast, the glassy state can be reached and no ice crystals are formed. However, when rewarming, ice crystals are easily generated due to recrystallization. In order to avoid damage to biological samples by ice crystals, the common methods used today are adding CPAs, changing the cooling rate, etc. CPAs can be classified according to their source as natural CPAs (e.g. antifreeze proteins (AFPs)), synthetic CPAs (e.g. polyvinyl alcohol (PVA))20. These CPAs are both permeable and non-permeable, and a combination of both is often used to reduce toxicity to cells. In addition, cryopreservation technology can destroy diseased tissues in situ and can treat conditions such as vascular lesions in the oral cavity21, primary tumors22, and cutaneous lesions23.”

Comment #4.slow cooling is inaccurate, please use equilibrium and/or quasiequlibrium cooling.

Response: We thank the reviewer for this suggestion. We would like to change “slow cooling”to“equilibrium cooling”in the text. Thank you! Please see page 4 of the revised manuscript, lines 93-98.

“The slow freezing mentioned in this paper is equilibrium freezing. This is because the chemical potential of the cytoplasm of the cell in the supercooled state is higher than that outside the cell, and in order to ensure the balance of the chemical potential inside and outside the cell so that the cell does not freeze, the cooling rate needs to be reduced, and for most cells, a slow cooling rate (e.g. 1°C/min) can significantly reduce the probability of intracellular freezing19,25.”

Comment #5. Along these lines, vitrification as defined in the manuscript only includes high concentration/high cooling rates. Again, organs/large tissues are vitrified using slow cooling rates, and note that water can be vitrified without any cryoprotectant.

Response: We thank the reviewer for this suggestion. It is absolutely correct that water can be vitrified without any cryoprotectant given high enough cooling/warming rates (e.g, > 105 ℃/min), and organs and tissues can be vitrified using slow cooling rate with high enough concentration of total CPAs (e.g, > 6 M). According to the available data, when vitrification is performed, if the cooling rate is slowed down, a large amount of cryoprotectants is added to increase the viscosity of the preservation solution to prevent the formation of nuclei in order to ensure that the vitrification state is achieved22,23. In order to ensure that no nuclei are formed during the vitrification process, the current method is to increase the cooling rate and/or add a large amount of cryoprotectants. Therefore, we would like to change “high concentration of cytotoxic CPAs and high cooling rates”to“high concentration of cytotoxic CPAs and/or high cooling rates”in line 117.

“line 117.”

Comment #6. Finally, the authors leave out some of the middle ground but successful techniques including interrupted slow cooling, two- or three-step methods, liquidus tracking, etc. 

Response: Thank you very much for reminding of these “middle ground” of cryopreservation techniques. Although many preservation methods have been proposed that make small or large modifications to the three conventional approaches, slow-freezing, vitrification, and hypothermia, to obtain the optimal outcomes for the specific biospecimens, the thermodynamic procedures, microenvironment conditions, and cell stresses are not far from one of these three approaches. As a result, we only discuss the molecular mechanisms of these major approaches in the paper. Nevertheless, we would like to explain this for these “middle ground” techniques in the paper. Please see page 6 of the revised manuscript, lines 142-163.

“In addition, there are some improved cryopreservation techniques, such as interrupted slow cooling, two- or three-step methods, liquidus tracking, etc. Interrupted cooling protocols and interrupted rapid cooling preserve mammalian specimens in two steps, first down to a critical intermediate temperature and then down to storage temperature with or without holding time, allowing the research of the type of cryoinjuries suffered by cells at different stages of cryopreservation. It has been shown that the cryoinjuries to cells in the interrupted cooling protocols stage is solute damage45,46, and the cryoinjuries to cells in the interrupted rapid cooling stage is ice crystal damage45. The two- and three-step methods are improved to protect the cells from the toxic effects of CPAs during vitrification. Different cells have different osmotic pressure tolerance limits and toxicity tolerance limits so that need to gradually load or unload the CPAs to reduce the toxicity of the CPAs to the cells47. The liquidus tracking is a method to reduce the concentration of CPAs and to shorten the time of exposure of cells to CPAs. This technique solves the problems of CPAs toxicity and ice crystal formation during vitrification48. The above three cryopreservation techniques are all improved methods for solute damage, ice damage, and toxicity of CPAs, and the molecular pathways of cryogenic damage to cells under cryogenic conditions are not mentioned. Since the metabolic states involved in the above three methods are similar to those of conventional cryopreservation techniques, we can classify the molecular pathways of cryoinjuries involved in cells in these three preservation techniques into the same category.”

Comment #7. Please ensure that all abbreviations are defined in the first instance. 

Response: We thank the reviewer for this suggestion. We are sorry for this error and have fixed it.

Comment #8. pH should be capitalized as such, Ca2+ should be superscripted throughout. 

Response: We thank the reviewer for this suggestion. We are sorry for this error and have fixed it.

Comment #9. Figure 4 is unclear. What are the different layers, lines, cell types? 

Response: We thank the reviewer for this suggestion. We have fixed it.

Comment #10. There is an enormous literature on sperm cryopreservation and antioxidants, and a more recent literature on somatic cell and tissues and antioxidants in cryopreservation. The authors should include a more thorough investigation of these applications and their relevance to the molecular mechanisms. Examples include ovarian cells, ovarian tissues, testicular tissues, and bull, sheep, etc sperm, and more. 

Response: We thank the reviewer for this suggestion. The main functions of antioxidants such as melatonin1, resveratrol2, and ascorbic acid3 are to reduce the oxidative stress of endoplasmic reticulum and mitochondria during cryopreservation of biological samples. The cryoinjuries induced by oxidative stress in cryopreservation is described in detail in Section 2, so it will not be described in detail here. It was found that the use of any antioxidant during the cryopreservation of ovarian tissue or sperm increased the preservation efficiency (increased tissue antioxidant capacity, improved DNA integrity, and reduced apoptosis)24-27, but the specific apoptotic pathways were not investigated, so it is not known whether the mechanism of antioxidant protection against cryogenic damage is consistent with our conclusion.

Round 2

Reviewer 3 Report

59: Icing is an atypical term for intracellular ice formation. Icing evokes ideas of cells/tissues covered in ice, not filled with ice…I would suggest changing it. 

59: “the main factors of icing” should probably be “the main results of icing” 

77: “has its own” should probably be “have unique”

106: Liquidus is not correct.

117: the use of the (R) symbol is not necessary unless the authors are obligated via manufacturing or usage contracts

130: with regard to freeze drying, the authors state that it is convenient, and safe and inexpensive. This requires some clarification as freeze drying equipment can be very expensive and challenging to operate. The citation is not as relevant at the authors suggest because only the nuclei were preserved by freeze drying. 

Lines 131-134: “This also indicates  that different species of mammalian specimens are preserved in different ways, and researchers need to further delineate the cryopreservation methods for different species of mammalian specimens.” Is unclear and should be rewritten. 

149: “improved methods for solute damage…” should probably be “improved methods to reduce solute damage…”

439: “The current methods for solve the cryoinjuries (apoptosis, necroptosis, and IRI)”  is not written well and needs revision.

Regarding the response to the ROS comment #10: the authors suggest that the molecular mechanisms are not known in the manuscripts shown. This is probably a good reason to include the paragraph that is written in the response simply to highlight the absence of these data. 

New text: Grammar needs attention in a number of places in the newly inserted text. It seems to not have been proofread as carefully. 

Figure legends: Grammar needs attention in these legends. They seem to not have been proofread as carefully as the rest of the document. 

Fig 2 and Fig 4: Is the set of lines indicated by light blue circles supposed to be a double bilayer, indicating the mitochondrial membrane? It would not really make sense if so, but if so it should be clarified as such, and either way it is very difficult to see as shown. 
